# Disinfectant, Soap or Probiotic Cleaning? Surface Microbiome Diversity and Biofilm Competitive Exclusion

**DOI:** 10.3390/microorganisms8111726

**Published:** 2020-11-04

**Authors:** Wendy Stone, Janke Tolmay, Keira Tucker, Gideon M. Wolfaardt

**Affiliations:** 1Environmental Microbiology Laboratory, Water Institute, Department of Microbiology, Stellenbosch University, 7602 Stellenbosch, South Africa; janketolmay@gmail.com (J.T.); 17771293@sun.ac.za (K.T.); gmw@sun.ac.za (G.M.W.); 2Environmental Microbiology Laboratory, Department of Chemistry and Biology, Ryerson University, Toronto, ON M5B 2K3, Canada

**Keywords:** probiotic diversity, cleaning regime, species competition, surface communities, biofilm competition

## Abstract

This study extends probiotic cleaning research to a built environment. Through an eight-month cleaning trial, we compared the effect of three cleaning products (disinfectant, plain soap, and a probiotic cleaner containing a patented *Bacillus* spore consortium), and tap water as the control, on the resident microbiome of three common hospital surfaces (linoleum, ceramic, and stainless steel). Pathogens, *Escherichia coli* and *Staphylococcus aureus,* were deposited and desiccated, and competitive exclusion was assessed for each microbiome. Cell survival was shown to be an incomplete tool for measuring microbial competitive exclusion. Biofilm competition offered a fuller understanding of competitive dynamics. A test for culturable cell survival showed that both plain soap and probiotic cleaner regimes established a surface microbiome that outcompeted the two pathogens. A different picture emerged when observing biofilms with a deposited and desiccated GFP-labeled pathogen, *Pseudomonas aeruginosa*. Competitive exclusion was again demonstrated. On surfaces cleaned with disinfectant the pathogen outcompeted the microbiomes. On surfaces cleaned with plain soap, the microbiomes outcompeted the pathogen. However, on surfaces cleaned with probiotic cleaner, despite the exponentially higher surface microbial loads, the microbiome did not completely outcompete the pathogen. Thus, the standard culturable cell test for survival on a surface confirmed the competitive advantage that is typically reported for probiotic cleaners. However, observation of competition in biofilms showed that the more diverse microbiome (according to alpha and beta indices) established on a surface cleaned with plain soap had a better competitive advantage than the monoculture established by the probiotic cleaner. Therefore, microbial diversity appears to be as critical to the competitive exclusion principle as cell numbers. The study showed that both plain soap and probiotic cleaner fostered competitive exclusion far more effectively than disinfectant. Probiotic cleaners with microbial diversity could be worth considering for hospital cleaning.

## 1. Introduction

The link between ecological diversity and environmental balance is evident at all levels of life. In humans and animals, an imbalance of pathogens causes disease. On a global scale, such imbalances lead to plagues and ecological collapse. Understanding the effects of pathogen proliferation can inform approaches to human health and can be applied to environmental health policy. Since human and ecological health are interconnected, not independent, good hygiene policy connects inwards with microbiome health and outwards with environmental health.

Hospitals are a particularly interesting biome, in terms of microbial diversity and homeostasis. Here, the built environment becomes a pathogen reservoir as a result of the heavy traffic of pathogen-laden patients and the liberal use of antibiotics [1,2,3]. Research has explored the spatial and temporal patterns of nosocomial infections [4,5] and the way hospitals try to control them with increasingly aggressive disinfection regimes and antimicrobial agents [6]. Since the advent of routine antimicrobial cleaning in the mid−20th century, attention has been focused on keeping hospitals as microbe free as possible. Antibiotics have lulled people into relying on medicine to rectify imbalances, hence the crisis of drug-resistant pathogens [7]. Too often today’s approach to ecology is a reaction to catastrophe rather than the stewardship and harnessing of natural systems.

Approximately 7% of patients in developed countries and 10% of patients in developing countries pick up nosocomial infections in general hospital wards [8], with some studies reporting as many as 15% of patients [9]. Developing countries are particularly susceptible because of HIV/AIDS, resource shortages, and poor education [10]. However, even with modern facilities and the highest conventional disinfection standards, hospitals in developed countries are not exempt [3]. Excessive and inefficient use of disinfectants, similar to excessive and inefficient use of antibiotics, may increase a hospital’s risk of becoming an incubator for resistant pathogens [6,11]. Antimicrobial surfaces are the next popular frontier in the battle against nosocomial pathogen survival [12], with similar implications. In contrast, a diverse bacterial ecology can prevent pathogen proliferation by increasing competition for nutrients and space, or by secreting secondary metabolites that confer a survival advantage [13]. When microbial cells are removed from a surface, the diversity is disturbed. If a pathogen moves in, then, as the competitive exclusion principle, Gause’s Law, states, the reduced competition may enable it to thrive and colonize the space [14].

This principle has leached into popular science as an awareness of the potential of probiotics, and has boosted the market for fermented foods such as kefir and kombucha [15]. In this study, we use the term “probiotic” to mean harnessing living organisms to promote health not just in the digestive tract but also in a built environment, and in our case specifically in a hospital. The health benefits of “dirt” have been discussed in both scientific and popular literature [16] and faecal transplants are increasingly being used to treat gastrointestinal disease [17]. Given these indications of both scientific and popular awareness and acceptance of probiotics, it should not be difficult to facilitate shifts in policy and management if we can produce empirical evidence that using probiotics in cleanliness regimes reduces the risk of pathogen transfer. This hypothesis has been tested [18], and applied to, for example, buildings and air-conditioning [19], indoor plants [20] and cleaning products [3]. International drives such as the Healthier Hospitals Initiative [21], Healthcare No Harm [22] and Practice Greenhealth [23] have been promoting ecological balance in the healthcare industry, motivating, in particular, more responsible use of disinfectants. 

In this study we assessed competitive exclusion with the aim of fostering a better understanding of ecology in hospital cleaning regimes. The resident microbiomes established after an eight-month cleaning regime (comparing disinfectant, plain soap, probiotic cleaner, and ambient tap water as a control) were assessed for their competitive interaction with introduced desiccated pathogens. Competitive exclusion was evaluated by observing cell survival and biofilm dynamics. The cleaning regimes were applied to samples of three kinds of surface, i.e., ceramic, linoleum, and stainless steel, that were selected to represent surfaces common in hospitals, and were placed both indoors and outdoors to evaluate the effect of containment and to make the results representative of the wider hospital environment. In this paper, we use the term ”resident microbiome” to refer to the microbiome that had become established on the surfaces after the eight months’ cleaning process, consisting of any of the original microbes remaining, microbes that landed on the surface between the weekly cleanings, and microbes deposited via the cleaning process (from the water, the probiotic cleaner and the cloths).

## 2. Materials and Methods 

We explored the competitive exclusion of microbiomes established on three different surfaces exposed to indoor and outdoor air microbiomes and subjected, for eight months, to distinct cleaning regimes, using three cleaning products and a control. First, we measured the culturable survival of pathogens desiccated on these surfaces, and then the competitive exclusion during the biofilm growth of the deposited pathogens, at physiological temperatures.

### 2.1. Strain Acquisition, Characterization and Maintenance

Clinical *Escherichia coli* and *Staphylococcus aureus* strains (Stellenbosch University Medical Microbiology, Tygerberg Hospital, Western Cape, South Africa) were inoculated in tryptic soy broth (3 g·L^−1^, 250 rpm, 37 °C, 24 h) within three days of isolation from a human host, and freeze cultures were prepared for storage after two passages (40% v·v^−1^ glycerol, −80 °C). All strains were inoculated directly from freeze culture stocks and passaged twice prior to experimental inoculation onto surfaces, after eight months of surface cleaning (3 g·L^−1^ tryptic soy broth, 250 rpm, 37 °C, 24 h). The clinical strains were not maintained as bench cultures at ambient temperatures, to prevent adaptation to laboratory conditions. Strains were characterized by plotting growth curves (3 g·L^−1^ tryptic soy broth, 37 °C, 250 rpm, 45 h). MacConkey agar (Sigma-Aldrich, Cape Town, South Africa) was used for the identification of *E. coli*. Mannitol salt agar (Sigma-Aldrich, Cape Town, South Africa) was used for the identification of *S. aureus*. 

### 2.2. Cleaning Regime and Surface Selection

The resident microbiomes on three surface materials were subjected to four cleaning regimes (Table 1), placed indoors and outdoors, informed by local hospital cleaning protocols and surface materials. The established impact of the air microbiome on indoor health, often studied in the context of “sick building syndrome” [24,25], motivated the inclusion of both indoor and outdoor replicates. In this study, the indoor blocks were arranged in a soil sciences laboratory, selected as a clean but non-clinical built environment. They were arranged on the floor, under the benches used by students. The outdoor samples were on a roof, situated near trees, exposed to sunshine, wind, and wildlife (birds and insects). 

Each treatment, surface, and placement combination was assessed in triplicate, with 72 control samples and 72 experimental samples. For all culturing and biofilm studies, small blocks of ceramic (3 × 3 cm standard sized, smooth, glazed bathroom tiles), linoleum (1.5 × 1.5 cm) and stainless steel (1.5 × 1.5 cm) were arranged randomly, indoors (in the Stellenbosch University Soil Sciences laboratory) and outdoors (on an exposed roof). For DNA extraction and molecular analysis of the resident surface microbiome diversity, large blocks of linoleum (45 × 30 cm) were arranged and cleaned according to the same four regimes (larger blocks were used to increase DNA concentrations and were placed only indoors because of logistical limitations). Four indoor cloths and four outdoor cloths were dedicated to each treatment (disinfectant, plain soap, probiotic cleaner and control, Table 1), air-dried between washes and stored separately. The probiotic cleaner we used contained a consortium of species from only one genus, *Bacillus* spores. Probiotic cleaners generally contain only *Bacillus* spores, with some brands claiming, at most, “a variety of species or strains”. The blocks to be treated with probiotic cleaner were separated from the other blocks by at least 20 cm to prevent cross-contamination. Surfaces were cleaned twice a week for eight months, allowing a resident microbiome to become well established. The surfaces were thoroughly rinsed of disinfectant or soap after each cleaning, to prevent the residue having an impact on cell survival.

### 2.3. Cleaning Regime Impact on Community Dynamics

#### 2.3.1. Resident Surface Community Quantification

Four hours after the final cleaning at the end of the eight-month experiment, all the blocks were transferred to the laboratory, where all subsequent work was done. The small blocks were vortexed to assess the baseline cell concentrations of the total cells (tryptic soy agar, 3 g·L^−1^ tryptic soy broth, 15 g·L^−1^ agar), *E. coli* (MacConkey agar) and *S. aureus* (mannitol salt agar) of the resident microbiome established at the end of the cleaning regimes. Baselines for each treatment, surface, and placement (indoors or outdoors) were assessed in triplicate. Each block was submerged face down in 5 mL sterile physiological saline solution (9 g·L^−1^ NaCl, 50 mL conical tube), incubated at room temperature for ten minutes, and vortexed for one minute [26], as well as in-study optimization. The cell suspension was quantified via dilution series (saline solution) on tryptic soy agar, MacConkey agar, and mannitol salt agar. Squares (1.5 × 1.5 cm) were cut from each cloth and the cell concentration similarly assessed in triplicate. 

#### 2.3.2. Resident Surface Community Diversity 

Community diversity was assessed for each cleaning regime in quadruplicate, by isolating the DNA of the resident microbiome from the surface of the large linoleum blocks. These blocks were transferred to the lab individually, and each surface sampled with a sterile gauze swab (5 × 5 cm, double layered) moistened with 1 mL sterile saline solution. Medium pressure was applied over the entire surface, first wiping horizontally, and then turning the swab over and wiping vertically. The swabs were placed in separate sterile 50 mL conical centrifuge tubes containing 20 mL phosphate buffered saline (pH 7.4) with 0.1% (*v*/*v*) Tween^®^ 20 detergent (PBST). Various cell disruption methods (vortexing, sonication, and combinations of these methods, as well as increasing intervals of both) were evaluated for separation of the cells from the swabs. The method finally selected was as follows. The centrifuge tubes were vortexed (1 min), the swabs were removed from the tubes and excess buffer was wrung out into the tube. Then, the swabs were placed in separate Petri dishes and an extra 5 mL PBST was added. Then, the swabs were rubbed and wrung out into the centrifuge tubes, the way one would wring out a dishcloth (this was established as a critical step for improving cell removal from the swabs during this study), bringing the PBST in each centrifuge tube close to a total of 25 mL. The swabs were then discarded. The tubes were centrifuged (5000 × g for 15 min), the supernatant carefully decanted, and the cell pellet resuspended in 500 μL saline solution.

The concentrations of the cell pellets were quantified by a dilution series on tryptic soy agar. Various DNA extraction methods (phenol/chloroform/isoamyl alcohol, Zymo kits or Qiagen kits) were evaluated prior to selection. The method finally chosen was extraction using the QIAamp^®^ DNA Micro Kit (Qiagen, Germany). The manufacturer’s protocol for blood samples was followed, with adjustments for low cell concentrations and high proportions of particulate matter to cells. The maximum initial cell suspension volume (100 μL) was used, adding carrier DNA to all samples and doubling the water bath incubation time (20 min) for enhanced cell lysis. When transferring the lysate to the column, care was taken to avoid the precipitated dirt, as earlier optimization demonstrated interference with the extraction procedure. The DNA was eluted in 25 μL distilled H_2_O and stored at 4 °C for subsequent molecular analyses within two days.

Automated ribosomal intergenic spatial analysis (ARISA) was commercially performed by Sporatec, Stellenbosch, South Africa. Eubacterial specific primers, ITSReub and FAM (carboxy-fluorescein)-labeled ITSF, were used to determine bacterial diversity [27]. The fungal diversity was determined with fungal primers ITS4 and FAM-labeled ITS5 [28]. The study focused on the bacterial microbiome, but genetic analysis included fungi for a fuller picture. ARISA sequencing and data processing methods are described in Section A.1.

#### 2.3.3. Competitive Exclusion: Cell Survival

Clinical *E. coli* and *S. aureus* strains were desiccated on blocks of each cleaning treatment, surface type, and placement (indoor or outdoor), and the competitive impact of each of the resident microbiomes on the survival of the desiccated pathogens was assessed. After eight months of cleaning, the small blocks were transferred to the laboratory, and the clinical pathogens were deposited to assess competitive exclusion.

A preliminary screening determined the most effective desiccation and disruption methods for consistent cell recovery (ambient vs. laminar flow desiccation, saline vs. tap water suspension, vortexing vs. sonication, optimizing cell concentrations, disruption rigor, and desiccation periods). The method selected was as follows: *E. coli* and *S. aureus* were grown well into their stationary phase, in triplicate (3 g·L^−1^ tryptic soy broth, 37 °C, 250 rpm, 36 h agitation, 12 h standing). Cells were harvested by centrifugation (2× 2 mL, 7500 × *g*, 3 min), washed in 2 mL sterile tap water, centrifuged (7500× *g*, 3 min) and re-suspended in 2 mL saline solution. Two separate 50 μL droplets, one of each strain suspension (*E. coli* and *S. aureus,* 10^6^ cfu/mL), were deposited next to each other on each small block (four treatments, three surface materials, and two placements). The blocks were air dried under laminar flow (ambient temperature, ±24 °C) for two and a half to three hours, and subsequently stored in Petri dishes on a benchtop (at ambient temperatures, 24 h). Un-inoculated control blocks were subjected to similar drying, for subtraction of false positives to compensate for some of the limitations associated with culturing methods. Post-desiccation viable cell survival was evaluated with a dilution series on tryptic soy agar, MacConkey agar, and mannitol salt agar. Each block was placed face down in 5 mL saline solution in a 50 mL conical tube, incubated at ambient temperature for ten minutes, and vortexed (1 min) before dilutions. The percentage cell survival was compared for *E. coli* (MacConkey agar) and *S. aureus* (mannitol salt agar) according to Equation (1): (1)$$\frac{\left[x\right]-{\left[x\right]}_{O}}{\left[I\right]}*100=\%\;cell\;suvival$$
where [x] is the final cell concentration per block (cfu·block^−1^), [x]_O_ is the final cell concentration per un-inoculated block (cfu·block^−1^), and [I] is the inoculum concentration per block (cfu·50 μL^−1^).

#### 2.3.4. Competitive Exclusion: Biofilm Proliferation

Indoor linoleum blocks reserved for this experiment were subjected to each cleaning regime for eight months, and then removed to the laboratory. The pathogen *Pseudomonas aeruginosa* (PAO1-GFP) was grown under physiological conditions and deposited in saline (representing sputum) on the blocks and desiccated overnight. Competitive population dynamics were assessed with subsequent biofilm regrowth of the linoleum block microbiome and the desiccated pathogen, at a temperature simulating human physiology and favoring the adaptation of the pathogen (37 °C). The total microbiome was stained with a universal fluorescent dye, and the competitive survival of the deposited pathogen was visualized with GFP using fluorescence microscopy.

The competitive exclusion of pathogenic biofilms by the resident microbiomes of the four indoor cleaning regimes was evaluated with confocal microscopy. A genome-integrated GFP-labeled *P. aeruginosa* strain of human origin (PAO1-GFP, transformed for previous studies at Ryerson University, Toronto, ON, Canada), was maintained (passaged at least twice), grown (at 37 °C), harvested and washed as explained above (Section 2.3.3). Small (5 × 5 mm) sections of the linoleum blocks, which had been subjected to each cleaning regime indoors for eight months, were inoculated with 50 μL droplets of the pathogen (10^5^ cfu/mL, pre-study optimization), and desiccated under laminar flow in open Petri dishes, for three hours. Indoor linoleum blocks were selected for the biofilm study because they could be cut small enough to fit into flow cells, and the effect of the cleaning regime was the only variable assessed. Un-inoculated controls were also included, in triplicate for each treatment. The flow cells (three channels per cell, designed to hold the small linoleum pieces) were sterilized by soaking in sodium hypochlorite disinfectant solution (500 mL, 5% m·v^−1^, 30 min), and washing thoroughly in a consecutive series of six sterile distilled H_2_O beakers (500 mL/beaker). Glass cover slips were briefly flame sterilized with ethanol, and small (3 cm) sections of tubing and connectors were autoclave sterilized. The inoculated and un-inoculated blocks were transferred into the respective channels, two blocks per channel, the coverslips glued onto the channels with marine silicone sealant (Bostik), and the tubing sections with connectors glued into each channel. The flow cells were left under laminar flow to cure for 24 h.

Flow reactor systems were assembled and sterilized in a 37 °C incubator room prior to connecting the flow cells. Silicone tubing (inner diameter 1.575 mm) connected the medium reservoir to the waste receptacle, passing liquid through a 15 mL glass bubble trap via a Watson Marlow peristaltic pump. The system was sterilized with a continuous flow of a commercial hypochlorite disinfectant solution (0.7% m·v^−1^, 4 h, 15 mL·h^−1^), and washed with sterile distilled H_2_O (12 h). Growth medium (3 g·L^−1^ tryptic soy broth) was continuously passed through the system (3 h) directly prior to connecting the flow cells. The flow was then stopped and the cured flow cells containing linoleum pieces were connected after the bubble traps, ensuring aseptic transfer and connection. The channels were filled with tryptic soy broth and the flow stopped for 1 h to allow cell surface attachment in the liquid environment. The flow was, then, restarted and maintained for 48 h. The flow cells were disconnected and analyzed with confocal microscopy, comparing the proportion of GFP-labeled *P. aeruginosa* cells to the total biome, stained with Hoecsht 33342, a universal blue fluorescent bisbenzimide DNA-binding stain. The channels were stained for half an hour with 3 mL Hoecsht stain per channel, diluted in saline solution (5 μg/mL), and rinsed with saline solution. A Zeiss LSM 780 with ELYRA PS1 confocal microscope with an alpha Plan-Apochromat 100×/1.46 Oil DIC objective was used, according to de Waal et al. [29]. Confocal imaging and image processing are described in Section A.2.

### 2.4. Statistical Analyses

All means and standard deviations of triplicate values were determined in Excel and all graphs were generated in the Veusz scientific plotting application (https://github.com/veusz/veusz). Individual treatment effects relative to the controls were assessed in Excel with Student’s t tests for differences in independent means, with a confidence interval of 95% (*p* < 0.05). For competitive cell survival, a three-way analysis of variance (ANOVA) was done using the l m package in R (R Core Team, 2013) to test the effects of each treatment, surface, and placement on percentage survival, with a Fischer least significant difference (LSD) post hoc test. 

## 3. Results

Microbial diversity and competitive exclusion were assessed by exploring the interactions of the desiccated pathogens with the resident surface microbiomes. Resident microbiomes were established on surfaces exposed to distinct cleaning regimes (disinfectant, plain soap, probiotic cleaner, and tap water control), on three surface materials (ceramic, linoleum, and stainless steel), placed indoors and outdoors. After eight months of cleaning, the resident microbiome of each surface was assessed for cell concentration and diversity. Subsequently, the clinical pathogens were deposited on each surface and desiccated, and the competitive dynamics were quantified in terms of culturable cell survival and biofilm proliferation. Because surface-air interfaces are considered to be extreme environments [30] it is necessary to optimize all techniques at the low end of detection thresholds. 

### 3.1. Resident Populations: Impact of Cleaning Regimes

To understand the competitive survival of pathogens deposited and desiccated on a surface, we first needed to understand the resident surface microbiome of each eight-month cleaning regime. Therefore, we compared the cell concentrations (culturing) and the diversity (ARISA) of the resident microbiomes. The impact of a cleaning regime on competitive exclusion was the primary focus; however, we also compared the effects of the surface materials and of indoor and outdoor placement. 

Intuitively, total cell numbers followed an increasing trend, with the lowest cell numbers found for cleaning with disinfectant, followed by plain soap, the control, and the probiotic cleaner. Total cell numbers per block were consistently higher outdoors than indoors for all cleaning regimes except the probiotic cleaner (*p* < 0.05, Student’s *t*-test of independent means). Probiotic cleaning established a surface cell population per block that was exponentially higher than for the other cleaning regimes, with the probiotic loads being much the same for indoor and outdoor placement. Probiotic cell numbers were lower on stainless steel (10^4^ < 10^5^ cfu/block) than on linoleum and ceramic (both 10^5^ < 10^6^ cfu/block) (*p* < 0.05, Student’s *t*-test of independent means). The *E. coli* and *S. aureus* concentrations in the resident microbiome population on the surface materials were quantified as a baseline, for subtraction during inoculation and desiccation experiments. Baseline concentrations were sporadic, not higher than 10^1^ < 10^2^ cfu/block (Figure 1). The cloths, being rough, had higher total *E. coli* and *S. aureus* cell concentrations per square than any of the surface materials. The resident *E. coli* cell concentrations were higher than the *S. aureus* cell concentrations on the cloths, for both indoor and outdoor cleaning (*p* < 0.05, Student’s *t*-test for independent means).

The microbiome alpha diversity (the species diversity within a sample) on the indoor linoleum blocks followed an unexpected trend in response to the cleaning regimes. The exponential Shannon diversity index and the inverse Simpson index showed that the microbiome diversity was lowest on surfaces cleaned with tap water, higher on those cleaned with plain soap, and highest on those that were disinfected (Figure 2). As expected, the bacterial probiotic alpha diversity was also low, comparable to those cleaned with tap water. Both bacterial diversity and fungal diversity were limited by the probiotic cleaning, but the spread (standard deviation) of the fungal diversity indices was much wider than that of the bacterial diversity indices for this cleaning regime. 

The prokaryotic and eukaryotic beta diversity (differences in species diversity between ecosystems) also provided comparative information on the diversity of the surfaces, with tightly associated points on the graph representing lower diversity. Eukaryotic diversity was higher than prokaryotic diversity, as is evident in both the alpha (Figure 2) and beta (Figure 3) diversity plots. The beta diversity indices showed less resolution than the alpha diversity indices. All four cleaning regimes had distinct yet generally overlapping beta diversity plots, with only the probiotic cleaning regime indicating a notably tight and unique bacterial diversity footprint (Figure 3A). The predominance of morphologically uniform *Bacillus* spores was clearly visible on the linoleum surfaces, both indoor and outdoor, cleaned with probiotic cleaner (see Appendix B and Figure A1).

### 3.2. Pathogen Persistence: Impact of Cleaning Regimes

We used a multivariate ANOVA with Fischer LSD post hoc analysis to compare the competitive survival of clinical *E. coli* and *S. aureus* strains in the four cleaning regime microbiomes. These pathogens were deposited and desiccated on the surfaces (ceramic, linoleum, and stainless steel), indoors and outdoors, which had been cleaned for eight months with distinct regimes (control, plain soap, disinfectant, and probiotic cleaner). This made it possible to differentiate between the viable survival responses of the clinical strains, adapted to physiological conditions, and those of the resident surface microbiome, adapted to ambient conditions. The results suggested that surface type had an impact on the pathogens’ tolerance of desiccation at the air-surface interface. *E. coli* survived best when desiccated on ceramic and *S. aureus* on stainless steel. The desiccation persistence of both the deposited pathogens was most pronounced on the control and disinfected surfaces (the latter thoroughly rinsed of disinfectant residue). The plain soap and the probiotic cleaner both limited the survival of the pathogens more than the disinfectant and tap water controls (Figure 4). 

### 3.3. Biofilm Proliferation: Impact of Cleaning Regimes

The competitive survival results above were compared to competitive exclusion in a biofilm study, differentially visualized with fluorescence. *P. aeruginosa* showed no competitive advantage in the biofilms of surfaces cleaned with the tap water control. The resident microbiome outcompeted the pathogen entirely (Figure 5 and Figure 6A). The resident microbiome of the surfaces cleaned with plain soap also almost completely inhibited the competitive growth of *P. aeruginosa* (Figure 5 and Figure 6B). However, *P. aeruginosa* predominated on the biofilms established on the surfaces cleaned with disinfectant (Figure 5 and Figure 6C). Interestingly, there were distinct qualitative morphological trends among the treatments. Smaller cells and more extrapolymeric substances were evident in the plain soap and control samples (Figure 6A,B), and brighter, larger cells, which are likely to be monocultures, formed distinct chain structures in both the probiotic and disinfectant treatments (Figure 6C,D). The probiotic microbiome outcompeted the pathogen in almost all the imaging windows, with the lowest box-plot quartile showing the least variation (almost all the samples had zero *P. aeruginosa* competitive viability, Figure 5 and Figure 6D). However, four imaging windows spread across all three probiotic biological replicates were completely dominated by *P. aeruginosa* (Figure 6D). 

## 4. Discussion

### 4.1. Placement and Materials

As expected, outdoor placement resulted in higher baseline cell concentrations than indoors for all test conditions except those treated with the probiotic cleaner (Figure 1, *p* < 0.05, Student’s *t*-test of independent means). In the probiotic case, the cell numbers were between one and five orders of magnitude higher than the surfaces treated with tap water, plain soap, or disinfectant, and had a relatively consistent microbial load (Figure 1). Despite the inherent and unavoidable variation in environmental studies, the statistical significance throughout the study is encouraging. This suggests that the treatment effects (the load of probiotic spores and the elimination of the surface microbiome with disinfection) supersede the impact of natural environmental variation. However, the reproducibility of this study, similar to many applied studies, is fundamentally limited due to the wide variation of environmental factors. 

In the case of the resident microbiomes, measured as surface microbial load, stainless steel had a more negative effect outdoors than indoors on the efficacy of the probiotic cleaner (Figure 1, *p* < 0.05, Student’s *t*-test for independent means). The surface microbial load was 10-fold lower on outdoor stainless-steel surfaces. This may have been due to metal ions on stainless steel surfaces influencing surface attachment, particularly with the added stress of ambient UV exposure and wind outdoors. It has been shown that spores such as *Bacillus* have a net negative charge similar to the net negative surface charge of stainless steel [31]. These repulsive forces may influence the surface attachment of the probiotic spores, making these dried surface communities more sensitive to wind disturbance or UV stress than they would be on linoleum or ceramic.

Surface material plays a role in pathogen survival, and surface roughness is an important factor [32,33]. Cloths hosted a higher cellular concentration per square than any of the smooth surfaces (Figure 1), supporting the idea that surface roughness, and thus porosity, promotes cellular attachment [32], and also supporting studies warning that fibrous cleaning materials can act as a pathogen reservoir for recontamination [34].

Surface materials also had a species-specific impact on the competitive desiccation tolerance of the pathogens (Figure 4). *E. coli* was most persistent on ceramic surfaces, whereas the more desiccation-tolerant Gram-positive species, *S. aureus*, was most persistent on stainless steel. The ionic surface charge and reflective nature of stainless steel, which probably increases UV exposure, may have stimulated dormancy desiccation responses in *S. aureus*. Oxidative, radiation, and desiccation stress stimulate effective dormancy responses and thus may promote survival [35]. Therefore, the surface charge and reflective nature of stainless steel may stimulate a stress response during desiccation, promoting dormancy, and thus survival. 

### 4.2. Resident Microbiome

Cleaning with plain soap and disinfectant increased resident surface-microbiome diversity more than cleaning with water (Figure 2 and Figure 3) and cell concentrations were decreased (Figure 1). A study has shown that using disinfectant to clean can increase diversity because it removes the predominant species, increasing the relative abundance of species usually masked in richer environments by faster growers [36]. However, another study has shown the opposite, that disinfectant decreases microbial diversity [37]. Thus, the change in diversity may depend on the diversity of the pre-disinfection microbiome, the efficiency of the disinfection, and the underlying diversity of the populations masked by high cell numbers. In our study, beta diversity plots showed relatively weak distinction between treatments, only clearly highlighting the low prokaryotic diversity of probiotic surfaces (Figure 3A). Alpha diversity plots indicated lower diversity in both the control and probiotic samples, and the combination of the alpha and beta diversity plots indicated that probiotic cleaner markedly inhibited the total prokaryotic surface diversity (Figure 2 and Figure 3), despite exponentially higher microbial loads on the surfaces treated with probiotic cleaner (Figure 1).

### 4.3. Competitive Exclusion: Survival and Biofilm Growth

These microbiomes left on surfaces cleaned with the four cleaning regimes had distinct influences on the persistence of pathogens, measured by culturable survival (*E. coli* and *S. aureus*, Figure 4). However, a different story emerged in the competitive biofilms, after a clinical pathogen (*P. aeruginosa*) was desiccated on the surfaces, and the total surface microbiomes were grown with the desiccated pathogen under biofilm conditions (37 °C). 

The microbiomes on the surfaces cleaned with tap water promoted the quantifiable desiccation tolerance of clinical *E. coli* and *S. aureus* strains better than those on the surfaces cleaned with plain soap or probiotic cleaner (Figure 4), probably because of the protective particulate matter on these surfaces, which were visibly dirtier than those cleaned by the other methods. Scanning electron microscopy demonstrated the association of cells with particulate matter on control surfaces, such as soil, threads, and pollen (see Appendix B and Figure A1). This protective association with particulate matter is a commonly reported occurrence, especially for desiccation tolerant Gram-positive organisms [38]. However, in contrast to culturable cell survival, the desiccated *P. aeruginosa* clinical pathogen was completely outcompeted in a biofilm by the resident control surface microbiome (Figure 5 and Figure 6A), which had high cell numbers (Figure 1), supporting the competitive exclusion principle. 

The microbiomes on the surfaces cleaned with disinfectant also promoted the quantifiable desiccation survival of *E. coli* and *S. aureus* better than those on the surfaces cleaned with plain soap and probiotic cleaner (Figure 4), but unlike the situation on the control surfaces, this was probably due to low resident cell numbers, and thus low competitive pressure (Figure 1). This hypothesis was supported, as biofilms grown post-desiccation were completely dominated by the desiccated clinical pathogen *P. aeruginosa* (Figure 5 and Figure 6C). Scanning electron microscopy also revealed that the disinfectant had damaged the linoleum surface, which did not happen with the probiotic cleaner or plain soap, another reason to shift from heavy reliance on disinfectants (Appendix B and Figure A1).

Cleaning surfaces with plain soap did not limit the resident microbial load as much as the disinfectant (Figure 1), and the resident microbiome demonstrated more diversity than was the case with probiotic cleaner (Figure 2 and Figure 3). This microbiome limited both the culturable survival and the competitive biofilm growth of all the desiccated clinical pathogens (Figure 4, Figure 5 and Figure 6B). Cleaning with probiotic cleaner resulted in a resident microbiome cell concentration exponentially higher than was the case with the other three treatments (Figure 1), which limited the culturable survival of desiccated pathogens (Figure 4), as predicted by the competitive exclusion principle. However, the crux of these results is that, when grown under biofilm conditions after pathogen desiccation, the probiotic monoculture had a lower competitive advantage against the pathogen than the plain soap surface microbiomes (Figure 5 and Figure 6). Most of the samples were outcompeted by the probiotic cells (low box-plot quartile, Figure 5), again supporting the competitive exclusion principle, but there were a few localized areas across all three probiotic biofilms where *P. aeruginosa* outcompeted the probiotic spores (Figure 6D). 

The probiotic and the plain soap microbiomes both reduced the number of culturable pathogens that survived; however, when grown under biofilm conditions, the plain soap microbiome was more effective against *P. aeruginosa* than the probiotic microbiome. The combined alpha and beta diversity indices of the probiotic microbiome showed a drastic limitation in diversity as compared with the other three treatments. Although the control and the probiotic sample had a similar alpha diversity, the beta diversity revealed that diversity was more limited in the probiotic samples. According to the competitive exclusion principle, the exponentially higher probiotic cell numbers should outcompete the pathogen more effectively than the microbiome of the surfaces treated with plain soap. Our results showed the opposite, suggesting that both diversity and cell numbers were important for competitive exclusion. The natural diversity and the cell concentrations established by a plain soap cleaning regime proved to be the most effective cleaning strategy within the limitations of this study. Although diversity has been proven to contribute to the competitive exclusion principle [39], this has not leached into popular practice or industrial probiotic cleaner composition. Most probiotic cleaners consist of a patented *Bacillus* spore consortium or lactic acid bacteria [40,41,42,43]. This study raises practical awareness about diversity in the built environment. It has the potential to inform the diversity engineering in probiotic cleaners, and it raises questions about the quantitative extent to which diversity influences the competitive exclusion of opportunistic pathogens. For instance, PROBAC© has engineered diversity into their product [44], including photosynthetic bacteria, yeast, and lactic acid bacteria. However, we know of no studies exploring the quantitative thresholds of diversity that have a measurable impact on competitive exclusion. 

## 5. Conclusions

This study showed that, by inhibiting the resident microbiome on surfaces, disinfectants strongly promote the survival of deposited pathogens. The use of probiotic cleaners to limit nosocomial infections is gaining interest and commercial traction, with numerous reports of their efficacy in forming stable surface biofilms, and thus excluding pathogens [3,45]. 

Our cell survival results support these studies. We found that plain soap and probiotic cleaner were equally effective in excluding pathogens (Figure 4). Our probiotic cleaner, like most such products, contained only a patented *Bacillus* spore consortium. Such a heavy predominance of one genus in a natural environment is rare. Studies have shown that *Bacillus* spores, even at high concentrations, are not an infection risk [46]. The primary benefit of probiotic cleaning systems is the external strains they introduce into hospitals, demonstrably lowering the antimicrobial resistance of surface microbiomes [9,42,47]. Nevertheless, there is little natural precedent for this skewed monoculture biomimicry, and the ecological impacts of the probiotic approach are not yet understood. 

We found that cleaning with plain soap promoted greater surface microbiome diversity than cleaning with the probiotic cleaner, which suggests that plain soap cleaning is better for promoting biofilm regrowth. To overcome the probiotic cleaner’s tendency to limit microbiome diversity, and to overcome antimicrobial resistance in hospitals, we suggest that diversity in probiotic cleaners should be investigated as a contributor to competitive exclusion. For general household cleaning, where antimicrobial resistance is not a factor, our study suggests it is best to use plain soaps, rather than disinfectants or probiotic cleaners. 

This study suggests a shift away from reliance on disinfectants, to plain or probiotic soaps in general hospital cleaning protocols. However, cleaning protocols are specific to national regulations, and hospital areas. Operating theatres must be disinfected. General waiting areas, bathrooms, walls, and floors all have distinct cleaning protocols, with unique disinfection frequencies. We selected a cleaning frequency to represent the general areas, walls, etc. This also reflects more general household and business cleaning regimes, widening the impact of the results. However, a more frequent cleaning regime might impact these conclusions. Thus, a controlled in situ study in general hospital areas, using in situ cleaning regimes, would add weight to these conclusions. We also conclude that a wider toolbox than culturable survival is necessary to assess the competitive exclusion of pathogens.

Our measurements of the culturable survival of desiccated pathogens after our eight-month cleaning regimes supported the competitive exclusion principle, i.e., the resident microbiome had outcompeted the pathogens. Both the plain soap and the probiotic cleaner fostered competitive exclusion far more effectively than the disinfectant. However, when we looked at biofilm growth to measure the competitive exclusion of desiccated pathogens by the resident surface microbiomes, the picture that emerged showed that both microbial numbers and diversity were important for competitive exclusion. Although cleaning with the probiotic cleaner resulted in exponentially higher surface microbial loads than cleaning with the plain soap, the biofilm competitive exclusion was lower for this monoculture probiotic application. Therefore, we conclude that microbial diversity is as important as cell numbers for competitive exclusion, which is an important consideration when assessing methods for improving the microbial ecological health of the built environment, especially a hospital. 

## Figures and Tables

**Figure 1 microorganisms-08-01726-f001:**
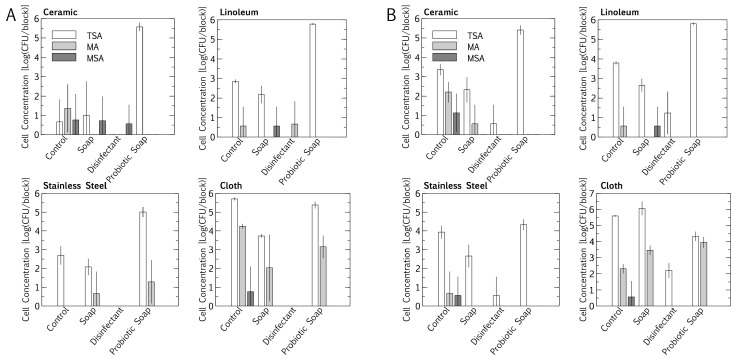
Impact of an eight-month cleaning regime (tap water control, plain soap, disinfectant, and probiotic cleaner) on microbial numbers of microbiomes on three surfaces, i.e., ceramic, linoleum and stainless steel, and on cloths used to wipe surfaces. Indoors (**A**) and outdoors (**B**) microbiomes are compared. The microbiome was assessed on tryptic soy agar (TSA, total), MacConkey agar (MA, *Escherichia coli*), mannitol salt agar (MSA, *Staphylococcus*
*aureus*). Results are expressed as mean ± SD of four swab samples.

**Figure 2 microorganisms-08-01726-f002:**
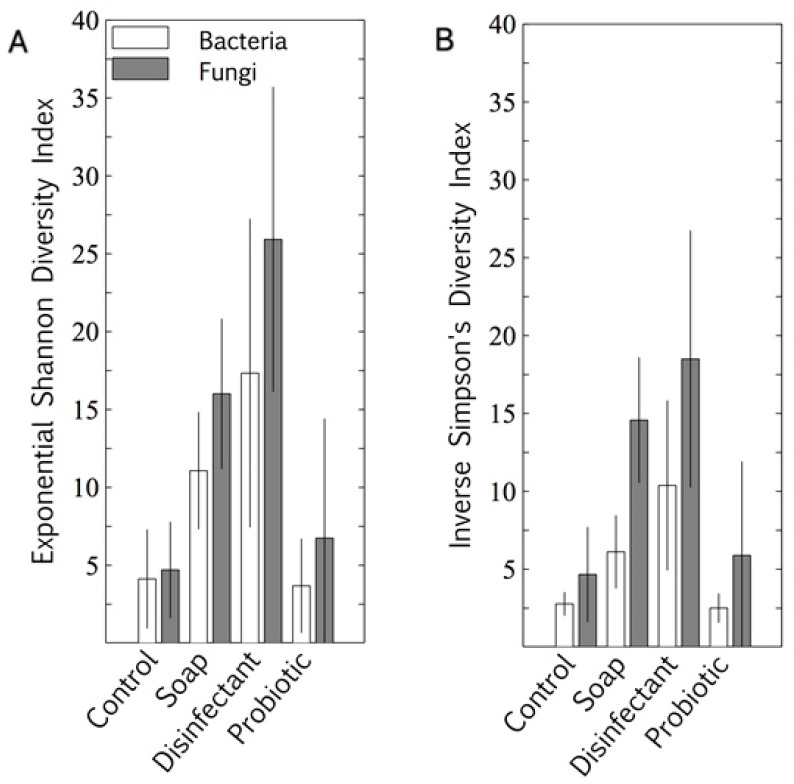
Bacterial and fungal alpha diversity indices of four cleaning regimes on indoor linoleum blocks, quantified as exponential Shannon diversity index (**A**) and inverse Simpson diversity index (**B**). Results are expressed as mean ± SD of quadruplicate swab samples.

**Figure 3 microorganisms-08-01726-f003:**
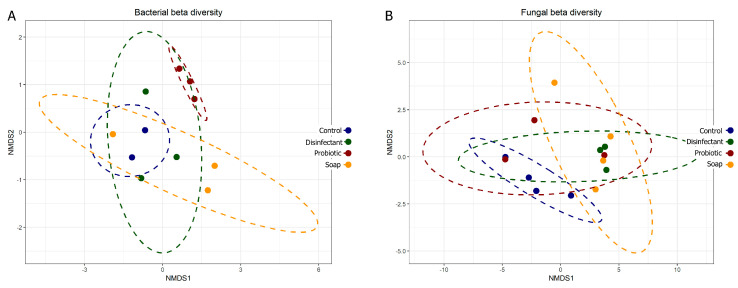
Bacterial (**A**) and fungal (**B**) beta diversity plots comparing community composition impacts of four cleaning regimes on indoor linoleum blocks. Each dot represents one of four swabs per cleaning regime, each plotted separately.

**Figure 4 microorganisms-08-01726-f004:**
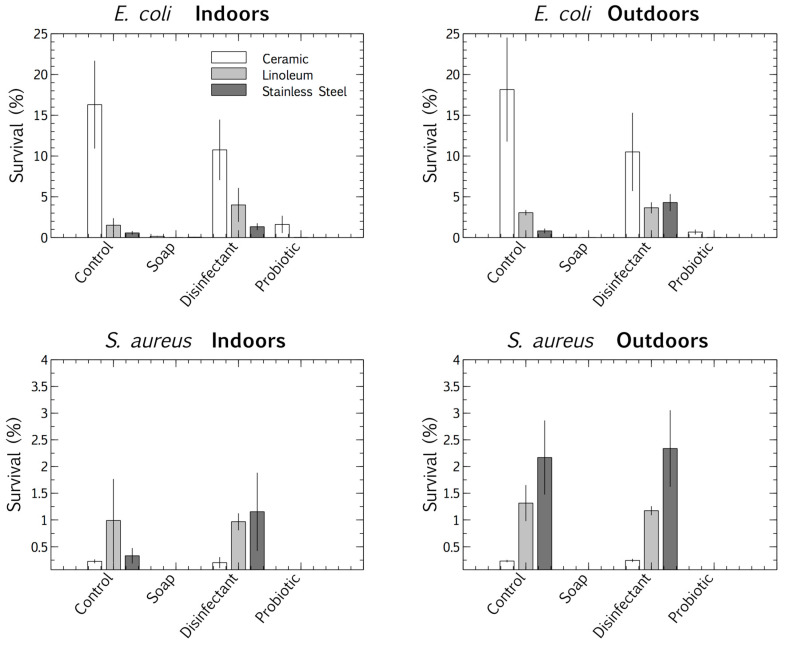
Percentage survival of *E. coli* and *S. aureus* after desiccation, deposited on three surfaces (ceramic, linoleum and stainless steel) indoors and outdoors, each treated with four cleaning regimes (tap water, plain soap, disinfectant, and probiotic cleaner). Results are expressed as mean ± SD of triplicate samples.

**Figure 5 microorganisms-08-01726-f005:**
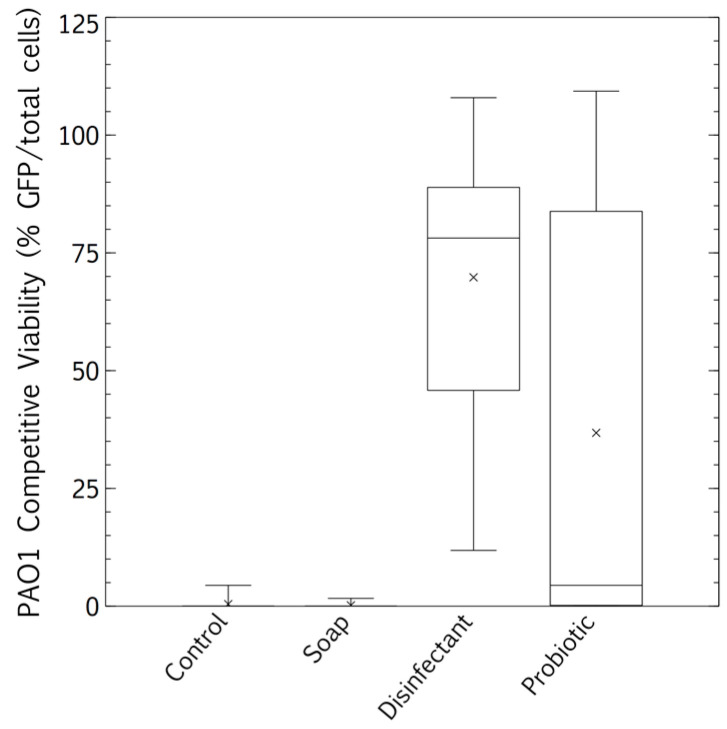
Biofilm proliferation of *Pseudomonas aeruginosa*-GFP in competition with the resident surface microbiome assessed between cleaning regimes (tap water, plain soap, disinfectant, and probiotic cleaner) using confocal microscopy. Post-desiccation competitive biofilm growth was calculated as % pathogen (green, *P. aeruginosa*) over the total microbiome (blue, Hoescht 537). Box and whisker plots represent averages of three flow cells, three images per flow cell (nine images per treatment, three biological replicates, all images at least 1 cm distant from each other).

**Figure 6 microorganisms-08-01726-f006:**
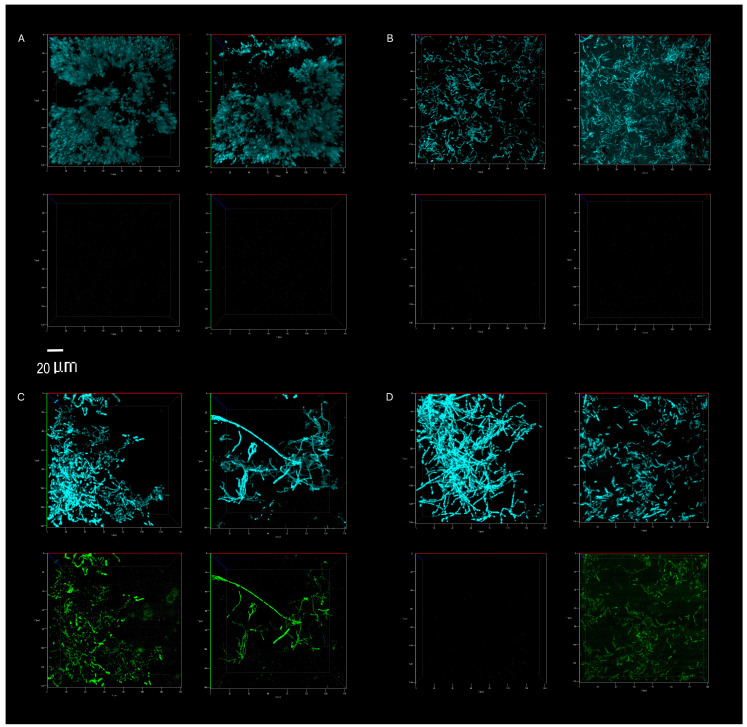
Direct visualization, using confocal microscopy, of the total microbiome (top panel of each image, blue) and *P. aeruginosa* (bottom panel of each image, green). Abundance on indoor linoleum squares was compared for tap water control (**A**), plain soap (**B**), disinfectant (**C**) and probiotic cleaner (**D**) treatments. The left and right sections of each panel (**A**–**D**) represent biological replicates, images of two separate flow cells per treatment. The 20 μm scale bar applies to all 16 panels.

**Table 1 microorganisms-08-01726-t001:** Cleaning products and procedures based on local hospital regimes.

Treatment	Product	Dilution	Procedure
Control	Tap water (Cold tap)	-	Rinse cloth, wring well, wipe once.
Plain soap	EarthSap biodegradable soap. Active ingredients: saponified vegetable extract, essential oils, natural gum	250 mL·500 mL^−1^	Wipe with cloth soaked in soap, rinse cloth thoroughly, wipe (×3).
Disinfectant	Jik bleach. Active ingredient: 3.5% m/v sodium hypochlorite	50 mL·500 mL^−1^	Wipe with disinfectant, soak for 5 min. Rinse cloth thoroughly, wipe (×3).
Probiotic cleaner	Bacterrorist non-toxic all-purpose cleaner (8.6 × 10^7^ colony forming units (cfu)·mL^−1^ *Bacillus* spores, analyzed in-study)	Undiluted	Spray directly onto surface, wipe with damp cloth.

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
