# Peer review of "Disinfectant, Soap or Probiotic Cleaning? Surface Microbiome Diversity and Biofilm Competitive Exclusion"

_microorganisms, 2020, doi:10.3390/microorganisms8111726_

Round 1
Reviewer 1 Report
In this manuscript, Stone et al., want to investigate the competitive exclusion of pathogens by the resident microbiomes of three kinds of surface after eight-month cleaning regimes. The topic of this work is current and the methods used are well designed. The topic of this work is important to find methods able to improve the hospital cleaning reducing the impact of nosocomial infections. demonstrate that probiotic, as stated in the definition, should be administered in adequate amounts. The review of literature, statement of the problem and framing of the study aims are clear. The main results showing that both microbial numbers and diversity are important for competitive exclusion are quite interesting even if it is not a new discovery. It is in fact known how, in ecosystems, the diversity and number of species is fundamental to avoid the prevalence of opportunistic pathogens. I would suggest to the authors to better argue the purpose of the work, especially to explain why a cleansing trial is carried out for eight months with three different methods: in my opinion, even prolonged treatment is unlikely to create a stable and characteristic microbiome, which can be influenced by many factors (temperature, exposure to air, presence of personnel, etc.). It is therefore difficult to create a standardized environment from which to obtain comparable results. Can the authors give more details on this?
Author Response
Reviewer Commentary
In this manuscript, Stone et al., want to investigate the competitive exclusion of pathogens by the resident microbiomes of three kinds of surface after eight-month cleaning regimes. The topic of this work is current and the methods used are well designed. The topic of this work is important to find methods able to improve the hospital cleaning reducing the impact of nosocomial infections. demonstrate that probiotic, as stated in the definition, should be administered in adequate amounts. The review of literature, statement of the problem and framing of the study aims are clear. The main results showing that both microbial numbers and diversity are important for competitive exclusion are quite interesting even if it is not a new discovery. It is in fact known how, in ecosystems, the diversity and number of species is fundamental to avoid the prevalence of opportunistic pathogens. I would suggest to the authors to better argue the purpose of the work, especially to explain why a cleansing trial is carried out for eight months with three different methods: in my opinion, even prolonged treatment is unlikely to create a stable and characteristic microbiome, which can be influenced by many factors (temperature, exposure to air, presence of personnel, etc.). It is therefore difficult to create a standardized environment from which to obtain comparable results. Can the authors give more details on this?
Author Response
We, the authors, would like to thank the reviewer for the positive response to the manuscript, as well as the astute in-depth questions about (1) the novelty of the work, and (2) the reproducibility.
Both considerations generated extensive discussion amongst us during the design and trial of the study. We are grateful for the nuance in the reviewer’s observations, and the opportunity to discuss these.
NOVELTY
Regarding the novelty of the work, we fully agree that there is data on diversity within the competitive exclusion principle and the inhibition of opportunistic pathogens [1, 2, and many more]. Nevertheless, we would like to make two arguments to motivate the necessity of this study, and our contribution within the broad umbrella of that principle.
1. This ecological principle is well-known within ecological spheres, but has not leached widely into applied microbiology sectors, nor popular consciousness. This is evidenced in the fact that, despite this ecological data, all of the probiotic cleaners we are aware of do not consider diversity a critical element. Ie. a. Counterculture uses exclusively lactic acid bacteria https://countercultureclean.com/products/baby-got-bac-2-pack b. The brand in our study uses Bacillus spores
c. Many other brands also use Bacillus spores, ie. Probio (https://www.probioclean.com/faq/), Mrs. Martin’s Probiotic Cleaners (https://mrsmartins.co.za/product/mrs-martins-probiotic-surface-soap-500mlaluminium/), and many others. d. The following patent suggests that these Bacillus spores are the preferred mix, and many companies are likely to stick to such patented mixes or something similar, to prevent having to prove safety of their product. https://patents.google.com/patent/CA2983075A1/en
e. Probac uses a mix of yeast, photosynthetic bacteria and lactic acid bacteria. However, we know of no studies considering quantifying the level of diversity needed to promote competitive exclusion. We suspect that this mix might have the same limitations as Bacillus spores (although, if there is data suggesting that this kind of engineered diversity quantitatively improves competitive exclusion, we’d be eager to revoke that suspicion!). https://www.probac.co.za/pages/the-probac-story
2. In current news, the international response to COVID-19 has been an indiscriminate shift towards disinfectant usage. We have seen very few voices raised in concern regarding the potential impact on opportunistic pathogen proliferation, and ecological balance. The natural diversity of the surface microbiome was not a critical decision-making factor in the past months, suggesting this knowledge is not in the forefront of our audiences’ minds. Thus, we believe an article like this one, which takes the ecological competitive exclusion principle and tests it in a very practical setting, may stimulate cross-pollination of these ideas between ecology and applied microbiology. We are hoping that an audience in the cleaning sector or hospital sector might gain a glimpse of this ecological principle, to inform their thinking. In addition, for any idea to leach into popular consciousness, the weight of repeated demonstrations in various applied contexts is as valuable as novelty. Thus, we believe that our final aim, which is crossing the bridge between academia and industry, is met within this article. 3. The true novelty of this study is the demonstration that the tool often used to measure competitive exclusion (culturing, survival) does not give us the complete picture. Here, we demonstrate that adding biofilm competition to our toolbox affords us a bigger picture. In our case, this bigger picture challenged our assumed notion that probiotic cleaners are sufficient.
These factors together lead to an interesting next question: what level of engineered diversity is sufficient, for microbial exclusion? For instance, Probac includes yeast, photosynthetic bacteria and lactic acid bacteria. Would this be sufficient to improve the competitive exclusion?
REPRODUCIBILITY
Regarding the reproducibility of the work, we agree with the reviewer that there is a wide array of variables in the study. We are long used to biofilm work, where we may challenge a biofilm with antibiotics and find two entirely different responses on two different days, even under controlled laboratory experimental conditions.
For this reason, we ran this project first with an Honours student as a pilot phase, and then repeated it as a full research project, both indoors and outdoors. We have extensive experience in the variability of biofilms, but were encouraged that we found repeatable, statistically significant results.
The reviewer is well within reason to point out this risk of variability, and we believe this is actually an extra strength of the study. It was to our surprise that we still found statistically reproducible results with such notable sources of environmental variation. Indoors and outdoors, in multiple pilot and execution phases, the results were statistically significant.
We believe this even further supports and strengthens our conclusions. Based on our data, the effects of probiotic soaps are drastic enough to supersede even the general variation in biofilm and environmental studies. In addition, disinfectant (bleach) is so effective in eliminating the microbiome that the effects between treatment also supersede the inherent variation in biofilm and ecological studies.
Thus, we agree with this concern, and are grateful that it was raised. However, it is precisely the reproducibility of the data in a wildly variable setting that excites us about this study: the fact that the treatments did produce a surface microbiome that had reproducible results both indoors and outdoors, with all the inherent variation.
Nevertheless, we did include a paragraph reminding the audience of the limitations of reproducibility due to variation in environmental and applied studies like this one.
Manuscript edits: The novelty of the study has been edited in Lines 521-529, and Lines 551-560. The reproducibility, and inherent limitations in most applied and environmental studies, is acknowledged in the discussion: Lines 423-428.
1. Dillon, R. J., Vennard, C. T., Buckling, A., & Charnley, A. K. (2005). Diversity of locust gut bacteria protects against pathogen invasion. Ecology Letters, 8(12), 1291-1298.
2. Matos, A., Kerkhof, L., & Garland, J. L. (2005). Effects of microbial community diversity on the survival of Pseudomonas aeruginosa in the wheat rhizosphere. Microbial ecology, 49(2), 257-264.
Reviewer 2 Report
This well written and clearly presented study investigated the effects of different cleaning regimes on the surface microbiome. I have only one major query:
The experimental design used a regime of once a week cleaning. In the 'real world' this would be more like daily cleaning. Can the authors comment on this and how it might change the data. Daily cleaning might prevent organisms from gaining a foothold which does not apply with only once a week removal. This might drastically change the data.
I have several minor queries:
1) pg 3 line 109 - this sentence doesn't make sense - please rewrite
2) The probiotic cleaner is described as 'Bacillus' and the authors refer to this as a species several times over. 'Bacillus' is the genus not the species. Could the authors state what the species and strain actually is within the cleaner.
3) in figure 1, there is no explanation of 'TSA,MA and MSA' in the legend.
Author Response
Reviewer 2 Commentary
This well written and clearly presented study investigated the effects of different cleaning regimes on the surface microbiome. I have only one major query:
The experimental design used a regime of once a week cleaning. In the 'real world' this would be more like daily cleaning. Can the authors comment on this and how it might change the data. Daily cleaning might prevent organisms from gaining a foothold which does not apply with only once a week removal. This might drastically change the data.
I have several minor queries:
1) pg 3 line 109 - this sentence doesn't make sense - please rewrite
2) The probiotic cleaner is described as 'Bacillus' and the authors refer to this as a species several times over. 'Bacillus' is the genus not the species. Could the authors state what the species and strain actually is within the cleaner.
3) in figure 1, there is no explanation of 'TSA,MA and MSA' in the legend.
Authors’ Response
The authors would like to thank the reviewer for the positive response to the manuscript, as well as both the minor editorial improvements and the opportunity to discuss the queries to the major philosophical underpinnings of the study.
The authors are grateful for the point about the cleaning frequency. As with reviewer 1, we’d like to acknowledge that this is an astute point, as it stimulated much discussion amongst the authors during experimental design.
The reason we chose once-per-week cleaning was based on earlier tension in the design of our study, as well as discussions with a local hospital about cleaning regimes.
In discussions with local hospitals about the impact of this work, we found the need to specify some critical points:
1. There are a number of cleaning protocols (standard protocols) in hospitals, specific to designated areas and turnover. For instance, the cleaning procedure in an operating theatre and bathroom is far more frequent and stringent than the general waiting areas and corridors. Terminal cleaning of a room, when a patient is discharged, is more stringent than non-terminal cleaning. In addition, our local hospital has been environmentally advised to avoid over-use of bleach, and thus might clean every day, but disinfect general areas less frequently (floors under waiting chairs, walls, etc).
Additionally, we were originally unsure whether to target this study specifically to the hospital environment, or to household cleaning, where overuse of disinfectant is also rife. Household cleaning is generally weekly.
It is critical to note that we do not recommend adjusting the disinfection cleaning protocol in the operating theatres, or where patients of invasive procedures have exposed wounds. In order to broaden the application of this work to more general cleaning of households and businesses too, we went with the lowest cleaning frequency representative of the most general areas. We have written a section cautioning interpretation of the data, and recommending in situ studies of the same phenomenon in hospitals, with in situ cleaning protocols.
Also important to the design is that, for all competitive exclusion experiments, the pathogens were deposited and desiccated between 4 and 12 hrs after final cleaning. Thus, it was within 24 hrs of cleaning (which would be a daily regime). This should minimize the foothold risk mentioned by the reviewer.
Manuscript Edits:
1. Lines 106-109: We would like to thank the author for pointing out the lack of clarity. We weren’t quite sure what was unclear (from within the study it is sometimes challenging to understand the audience viewpoint). Thus, we simplified Line 109 and hope the rationale is clearer.
2. This is a crucial point and we are very grateful to the reviewer for pointing this out. It led to a correction, which was edited throughout the paper. Based on the reviewer’s comment, we contacted the company producing the product. We contacted them once during the initiation of the study, and they were, understandably, unwilling to give out patented information on the species mix. They clarified in this week’s communication that it is more than one Bacillus species.
We hold applied microbiology at the core of our values. Thus, the final aim of this work is to enter into conversation with industry and our local hospitals, including this local probiotic company who were amenable to conversation. Thus, although it would take me mere days to figure out the individual strains by sequencing, we are very reticent to sequence the mix and violate their patent confidentiality. Our relations with them are as important to us as academic publication. In addition, we believe the diversity data is enough to make the conclusions necessary to our work. Species-level identification would not add enough information to the manuscript to balance the risk of losing good relations with industry. We hope to continue the work with them, engineering diversity into commercial probiotics. However, we are grateful that the reviewer pointed out that we referred to it as a single species, when it is a patented Bacillus spore consortium. We have corrected it throughout. Lines: 14, 130, 532-533, 546-547
3. Line 340 and 341: added TSA, MSA and MA in the legend. We are grateful for the correction.
Reviewer 3 Report
This is an interesting study exploring the microbiome of hospital surfaces after a period using three different sanitization protocols.
Before a possible acceptance the authors should address the following items:
- Please concise the methods section and if needed use supplementary materials.
- You should address the limits of your study in the discussion section (i.e. the fact that you are unable to control environmental outdoor factors acting on your surface).
- The conclusion referred to the practical application of your findings in daily hospital sanitization protocol is needed.
- Throughout the manuscript there are several typos and sentence that may be rephrased.
Author Response
Reviewer 3 Commentary
This is an interesting study exploring the microbiome of hospital surfaces after a period using three different sanitization protocols.
Before a possible acceptance the authors should address the following items:
• Please concise the methods section and if needed use supplementary materials.
• You should address the limits of your study in the discussion section (i.e. the fact that you are unable to control environmental outdoor factors acting on your surface).
• The conclusion referred to the practical application of your findings in daily hospital sanitization protocol is needed.
• Throughout the manuscript there are several typos and sentence that may be rephrased.
Author Response
We would like to express our gratitude to the reviewer for their interest in the study. We are also deeply grateful for the comments to improve the work, and are confident that the reviewer contributions improved the manuscript significantly.
1. We agree and thank the reviewer for this suggestion. We have summarized the methods by moving two Materials sections to Appendix A. Line 207-208, Lines 287-288
2. This point was raised by two separate reviewers, which always indicates an astute comment. We did add a sentence pointing out the variation in environmental studies, and inviting future work to assess the reproducibility. As mentioned to reviewer 1 too, we saw reproducible results indoors and outdoors, and with multiple shorter unpublished iterations of the study. We were encouraged and similarly surprised by the statistically significant differences between treatments, despite the inherent variability in both environmental studies and biofilm studies. It suggests that the effects of the treatments (ie. the heavy load of probiotic spores and the elimination of the surface microbiome with disinfectant) supersede the inherent natural variation introduced by the environment. This was encouraging to us. However, we agree that all environmental and applied studies introduce necessary variation that makes reproducibility challenging. We have written a sentence acknowledging this. Line 426-430.
3. We agree with this point, thank you, and have edited it into the discussion. Line 561-570. We have made some cautious conclusions, and recommended in situ studies to confirm this work (we hope to be doing that work soon ourselves, in collaboration with local hospitals).
4. We thank the reviewer for this point. We have reread the manuscript and made some grammatical changes where we could find mistakes. We are passionate about this work, and thus paid a professional language editor to correct it for us (despite being first language speakers) to facilitate communication of the work to an audience. So we hope the mistakes were at a minimum.
Line 52: added “a”
Line 112: added “were”
Line 155: Table 1, explained “cfu”, also replaced “CFU” with “cfu” throughout (was not consistent)
Line 214: added “was” Line 222: added spaces between numbers and units
Line 258: added a space between number and unit
Line 285: added rinsing Line 363: added “morphologically uniform” for clarity
Line 393: added “s” Line 400: removed a misplaced comma
Line 404: added “probiotic”
Line 425: replaced “or” with a comma